# Influenza A Virus NS1 Protein Structural Flexibility Analysis According to Its Structural Polymorphism Using Computational Approaches

**DOI:** 10.3390/ijms23031805

**Published:** 2022-02-04

**Authors:** Sarah Naceri, Daniel Marc, Anne-Claude Camproux, Delphine Flatters

**Affiliations:** 1Université de Paris, CNRS, INSERM, Unité de Biologie Fonctionnelle et Adaptative, 75013 Paris, France; sarah.naceri@etu.u-paris.fr (S.N.); anne-claude.camproux@u-paris.fr (A.-C.C.); 2Equipe 3IMo, UMR1282 Infectiologie et Santé Publique, INRAE, 37380 Nouzilly, France; daniel.marc@inrae.fr; 3UMR1282, Infectiologie et Santé Publique, Université de Tours, 37000 Tours, France

**Keywords:** molecular dynamics, influenza a virus, non-structural protein 1 polymorphism

## Abstract

Influenza A viruses are highly contagious RNA viruses that cause respiratory tract infections in humans and animals. Their non-structural protein NS1, a homodimer of two 230-residue chains, is the main viral factor in counteracting the antiviral defenses of the host cell. Its RNA-binding domain is an obligate dimer that is connected to each of the two effector domains by a highly flexible unstructured linker region of ten amino acids. The flexibility of NS1 is a key property that allows its effector domains and its RNA binding domain to interact with several protein partners or RNAs. The three-dimensional structures of full-length NS1 dimers revealed that the effector domains could adopt three distinct conformations as regards their mutual interactions and their orientation relative to the RNA binding domain (closed, semi-open and open). The origin of this structural polymorphism is currently being investigated and several hypotheses are proposed, among which one posits that it is a strain-specific property. In the present study, we explored through computational molecular modeling the dynamic and flexibility properties of NS1 from three important influenza virus A strains belonging to three distinct subtypes (H1N1, H6N6, H5N1), for which at least one conformation is available in the Protein Data Bank. In order to verify whether NS1 is stable in three forms for the three strains, we constructed homology models if the corresponding forms were not available in the Protein Data Bank. Molecular dynamics simulations were performed in order to predict the stability over time of the three distinct sequence variants of NS1, in each of their three distinct conformations. Our results favor the co-existence of three stable structural forms, regardless of the strain, but also suggest that the length of the linker, along with the presence of specific amino acids, modulate the dynamic properties and the flexibility of NS1.

## 1. Introduction

Influenza A viruses (IAVs) are highly contagious RNA viruses that cause respiratory tract infections in humans and animals. Being responsible for epizootics in birds and several species of mammals, they can transmit to humans, either as sporadic cases of human infection with swine or avian viruses, or as pandemic viruses, which eventually will infect a large part of the human population. The two latest pandemic influenza A viruses (H3N2 in 1968 and H1N1 in 2009) have since been responsible for seasonal influenza for more than 50 years and twelve years, respectively. Annually, seasonal influenza causes three to five million severe cases and 290,000 to 650,000 deaths due to respiratory complications [1,2].

The eight-segment viral genome encodes more than 10 viral proteins, among which is the non-structural protein 1 (NS1), which is absent from the virion but highly expressed in the infected cells. NS1 plays an important role in countering the antiviral defenses of the infected cell, thereby helping the virus to escape the innate immune system [3,4]. For this reason, novel therapeutic strategies have recently been explored that antagonize NS1’s activities [5,6]. NS1 is known to bind non-specifically to double-stranded RNA (dsRNA) as well as to viral and cellular RNAs [7]. However, the effector domain is also proving to be an important therapeutic target given its multiple protein partners and its role in the structural polymorphism of NS1 [6,8,9].

NS1 is a homodimer of two 230-residue chains. It is comprised of two domains that are connected by an unstructured and highly flexible region of ten amino acids, which can be noted as a long or short linker (respectively referred as LL and SL in this study). The RNA-binding domain (RBD) is an obligate dimer involving residues 1–73 of each chain, arranged as three pairs of symmetrically positioned anti-parallel alpha helices. The two antiparallel helices 2 and 2’ form the RNA-binding interface, thanks to several basic amino acids including arginine 38 and lysine 41. The RBD binds several double-stranded RNAs (dsRNAs) as well as viral and cellular RNA, with no obvious sequence-specificity. The RBD has long been identified as an independent domain; it has been expressed as a recombinant protein and its structure has been established, both by crystallography and by nuclear magnetic resonance [10,11]. The different three-dimensional (3D) structures available in the Protein Data Bank (PDB) [12] indicate that this folding is well conserved across the diversity of influenza A virus strains. Residues 86–203 of each chain fold autonomously in an effector domain (ED), which is composed of three helices and seven beta-strands; the latter are organized into a broad sheet of antiparallel strands enveloping a long helix [13] and have been described as stable as a monomer or as a dimer. The different structures available on the ED dimer showed several dimerization interfaces, either via the helix 5 (residues 170–188) to form a helix–helix interface, or via a short strand (residues 88–91) to form a strand–strand interface [14]. According to the position of the two ED domains relative to the RBD dimer, three distinct conformations or forms (open, closed, semi-open) have been described in the literature [15,16,17]. Through this structural polymorphism, ED domains play an important role in the interaction with different protein partners [18]. These interactions are responsible for disrupting the defense function of the host cell [8,19].

The first full-length NS1 structure, published in 2008, was that from an H5N1 strain (pdb id: 3F5T) [20]. This structure corresponds to an open conformation, with each ED facing the RBD dimer through its short strand at position 88–91 while the surface of the long helix 5 is exposed to the solvent. Subsequently, two additional complete NS1 structures from two other strains, H6N6 and H1N1, were available in the PDB (pdb id: 4OPH in 2014 and 5NT2 in 2018, respectively) [15,21]. These two structures describe an intermediate form where both sides of the ED are more or less exposed and correspond to semi-open conformations. These 3D structures highlight a structural polymorphism and emphasize its conformational plasticity. The hypothesis of the co-existence of these three forms for the NS1 protein would explain the numerous possibilities of interactions with its partners [22]. However, each of the three full-length NS1s that was experimentally crystallized yielded a unique structure, giving no experimental support to the putative flexibility of NS1 in vivo. This lack of data leads to several hypotheses to explain this structural polymorphism including dependence on strain, the length of the RBD-ED linker or specific NS1 sequence residues. Additionally, the possibility that NS1 adopts these three forms regardless of the strain but conditional to physiological conditions and localization (nuclear or cytoplasmic) in the cell was also proposed [14].

In this study, we focused on the structural polymorphism of NS1 for three strains with full length structures available in the PDB. These are the PR8 strain, of the H1N1 subtype, and the H6N6 and H5N1 strains, which are avian strains. Structurally, the H5N1 protein is available with a short linker in an open form [20]. The H1N1 protein is characterized by a long linker and a semi-open form. This protein was resolved as a complex with a coiled coil domain of the TRIM25 protein and in the presence of a mutation at positions 187. In this complex, only the ED domains interact with TRIM25 (via sheets face). The authors show that the refolding of each ED is only mildly affected by the interaction compared to the free-form ED. However, the authors showed that the relative position of the two ED domains is conditioned by the presence of TRIM25 in a semi-open form [21,23]. The structure of the NS1 protein of H6N6 is characterized by a long linker and a semi-open form without any partner or mutation. In this computational study, in order to verify whether NS1 is stable in all three forms and for the three strains, we constructed one model by homology for each of the two complementary forms not known experimentally by crystallography for each strain studied. In a second step, molecular dynamics simulations were performed for three forms obtained on the three strains to study the stability of the structures relative to the form and the strain. The structural and dynamics study of NS1 and the characterization of RBD-ED movements allowed us to better understand the dynamic properties of NS1 in its different forms for different strains in order to develop a strain-independent therapy.

## 2. Results

### 2.1. NS1 Protein Properties Depending on Three Strains

#### 2.1.1. Strain Specific Properties of NS1

The multiple alignment of the NS1 protein sequences of three H1N1, H6N6 and H5N1 strains (Uniprot identifiers P03496, Q20NS3 and A5A5U1 respectively), which all belong to the A allele of NS1, emphasizes the high degree of conservation (Figure 1a). The RBD region is particularly well conserved especially at the level of the α2 helix, which forms the RNA-interaction surface (Figure 1b). Only a few differences are observed, especially for residues located in a loop or at the end of a helix (3, 22, 48, 70, 71, 75). The H5N1 NS1 differs by its characteristic deletion of residues 80–84 in the linker region (short linker, SL). Some variations are also observed in the ED: residues 103, 106 and 114 in connecting loops, residues 118, 127, 170, 178, 195 and 198 in secondary structures, as well as residues 205, 207, 214, 216, 218, 221 in the C-terminal region (residue numbering is that of the H1N1 sequence). The C-terminal extension is not considered in this study because of its disordered behavior.

#### 2.1.2. NS1 Structural Properties According to the Three Forms

The crystal structures of the H1N1, H6N6 and H5N1 variants (PDB codes 5NT2, 4OPH and 3F5T respectively) are representative of the semi-open (5NT2, 4OPH) and open (3F5T) conformations, while the closed conformation is represented by the structure (PDB 4OPA) of another variant of the H6N6 sequence, which is characterized by the engineered deletion of residues 80–84, along with the presence of a glutamic acid at position 71 [15,21].

In addition to its crystal structure corresponding to a given conformation, each sequence was fitted, through a homology modelling (HM) approach, into the crystal structure representing the two remaining conformations, resulting in a set of three conformations for each sequence. Thus for the H1N1 and H6N6 variants, one closed and one open conformation were generated by homology, using the corresponding crystal structures of the H6N6 variant in the closed (pdb id: 4OPA) or open (pdb id: 6OQE) conformation as templates. Of note, the two latter structures are those of the H6N6 variant harboring the 80–84 deletion, a deletion that we reversed when building the HM models. The closed and semi-open conformations of the H5N1 variant were modeled using as templates the corresponding crystal structures of the H6N6 (pdb id: 4OPA) and H1N1 (pdb id: 5NT2) variants, respectively. The three crystal structures were named XR (H1N1^XR^, H5N1^XR^, H6N6^XR^), as opposed to HM, for the six homology models (for more details, see Section 4).

In Figure 2, for each conformation, we compared the three structures through alignment and superimposition using the Pymol software.

The NS1 structures show a faithful superimposition in the closed (H1N1^HM^, H5N1^HM^, H6N6^HM^) and open forms (H1N1^HM^, H5N1^XR^, H6N6^HM^) (Figure 2a,c), regardless of the linker length. In the semi-open form, three strain conformations (H1N1^XR^, H5N1^HM^ and H6N6^XR^) show a different orientation of the ED with respect to the RBD, specifically for H1N1 (Figure 2b). These variable conformations may suggest that the semi-open form is an intermediate state between the closed and open forms. The H6N6^XR^ conformation is in semi-open form with an open tendency and the H1N1^XR^ conformation is in semi-open form with a closed tendency just like the H5N1^HM^ conformation, which was constructed by homology from the H1N1^XR^ structure as a template. The structures of the same three forms overlap very well at the level of the dimeric RBD domain. Depending on the form, the main structural differences are observed in the position of each ED domain relative to the RBD domain or relative to each other.

In order to characterize the orientation of these ED monomers for the three forms and three strains, we calculated three distances (see Section 4). The distances of the geometric center (see method) for each initial static structure between the two monomeric ED (d1) domains and then between each of the monomers (chains A and B) of the ED domain and the RBD dimer (d2 and d3) (Figure 3 and Appendix A) were calculated. These initial static structures correspond to time 0 nanosecond (t = 0 ns) and were subsequently used as the initial structure to run 150 ns molecular dynamics simulations. The results are shown in Figure 3 and detailed in Appendix A.

The distance (d1) between the two ED monomers A and B in the closed conformation is consistently close to 28 Å, while it reaches its maximal values (69–75 Å) in the open conformation (Figure 3a,c, Appendix A). This distance is more variable in the semi-open form depending on the strain (Figure 3b). The H5N1 strain shows the smallest d1 in the semi-open form, which can be explained by the fact that it is a model built on a semi-open template with a closed tendency (H1N1 template) but also with a short linker. This brings the ED monomers closer together with a distance of 43.3 Å. The H1N1 and H6N6 strains in the semi-open form with a closed and open tendency result in a d1 distance of 47 Å and 59.8 Å respectively (Figure 3e,f). These differences in distance for the same form can be related to the crystallographic structure of the two strains, particularly H1N1, which was crystallized in a complex with TRIM25 via the ED domain, which was constrained by these interactions. Despite the variations in the d1 distance observed in particular for the semi-open forms, we note that d1 characterizes well the three possible structural forms for the NS1 protein (Figure 3d–g). The distances d2 and d3 are more or less symmetrical regardless of the strain or form, being close to 40 Å in the closed form and between 34 and 39 Å in the open form. These distances show more variations in the case of the semi-open form with the largest values for H1N1^XR^ (46.6 Å and 49.5 Å) and shortest values for H5N1^HM^ (35.6 Å and 33.7 Å). The H5N1 form is the unique form with a short linker compared to H1N1 or H6N6 forms.

These results observed on the initial structures show characteristics that are specific to each crystallographic structure associated with a structural form of NS1. These characteristics are also found in the model structures depending on the template structure used. Moreover, these homology models are simulated in different forms that are not yet experimentally solved (by crystallography or NMR). Molecular dynamics simulations allow us to verify the stability of such models. In order to get rid of the properties of the template structures and to highlight the intrinsic structural properties linked to each strain, we performed molecular dynamics simulations on these 9 initial structures (i.e., the 3 crystal structures and the 6 models built by homology).

### 2.2. Dynamic Properties of NS1 Structures

Each initial NS1 structure (t = 0) (in three forms crossed with the three strains) were submitted as a starting structure to molecular dynamics (MD) simulations. This MD approach allows one to explore the conformational space of each structure and to study the structural and dynamic properties, intrinsic to the protein sequence. From simulations of 150 ns duration, 150,000 snapshots (one conformation per picosecond) were extracted and analyzed (Figure 4).

#### 2.2.1. Stability of NS1 Structures from the Three Strains in Different Forms

The C𝛼 Root Mean Square Deviation (RMSD) curves were calculated on the 150,000 snapshots of each of the nine initial structures (Figure 5 and Appendix A). The RMSD values (average and standard deviation) of each monomer are represented in Table 1 and the corresponding histograms in Appendix A.

This shows that the structures are very stable in the closed form regardless of the strain, especially for H1N1^HM^ with average RMSD values ranging from 2 to 5 Å (Table 1). For the open forms, the RMSD curves still show a plateau but with higher values ranging from 9 to 12 Å for the three strains. These results show that the protein undergoes greater conformational changes from its initial structure at t = 0 ns for the open form (Figure 4). It is confirmed that the semi-open form is more variable depending on the strain. The RMSD are around 10 Å for H1N1^XR^ and H5N1^HM^. These two structures move away from the starting conformation over time and RMSDs reach values similar to those of the open form. On the other hand, in H6N6^XR^, this semi-open form seems to be quite stable throughout the trajectory with an RMSD average value not exceeding 5 Å (Table 1, Appendix A). Of note, the initial semi-open structure of H1N1 and H5N1 was derived from a crystallographic structure complexed with TRIM25 and molecular dynamics are performed on the non-complexed NS1 protein. During the simulation, the initial structure moves away from its complexed conformation and shows the characteristics of the non-complexed semi-open form.

In order to identify if these conformational changes, inducing high RMSD values, are mainly due to movements of the ED domain observed between the beginning and the end of the simulation or to intrinsic deformations of the ED domain fold or of the RBD dimer fold, we calculated the RMSD-RBD dimer (RMSD^RBD^) and each ED domain separately.

The C𝛼 RMSD of the RBD dimer conformations during the simulation compared to its initial structure (t = 0 ns) was calculated (Appendix A). The RMSD values vary between 1.9 and 3.3 Å for all nine trajectories. This result shows that the intrinsic fold of the RBD dimer is very stable during the trajectories regardless of the sequence variant and of the initial conformation (Table 1, line RMSD RBD, Appendix A).

Similarly, we calculated the C𝛼 RMSD of each chain of the ED domain during the simulation relative to their conformation in the initial structure (Appendix A). The results show a very good stability of each monomer of the ED domain with RMSD average values not exceeding 2.5 Å (Table 1, lines RMSD ED_A and RMSD ED_B).

The relatively high RMSD values that we observed on the whole protein are therefore not due to a deformation of the 3D structure of these two ED chains or to a change in the RBD dimer fold (Appendix A). Rather, these mainly correspond to the motions of the two EDs relative to the RBD, as shown by the consistently high RMSD values reported in Table 1, line RMSD ED fit to RBD.

To quantify the movement of each monomeric ED allowed by its flexible linker region, we calculated the C𝛼 RMSD of the ED domain (including the linker region) relative to each other. We first verified that the presence or absence of the linker did not alter the RMSD curve by overlaying the results. The RMSD curves of the ED monomers with and without linker were superimposed, showing that the movement of the ED and the linker are not dissociated.

The H1N1^HM^ and H5N1^HM^ structures are very stable in the closed form, with RMSD values of 7.1 and 5.6, respectively. In contrast, the H6N6^HM^ strain has a RMSD of 14.7 Å. These high RMSD values are not due to the ED monomers moving away from each other (the closed form is well maintained) but to the linker moving, causing the dimeric ED to move horizontally away from its initial position (Figure 4). The semi-open form seems to be less stable compared to these two strains. It should be remembered that H1N1^XR^ (5NT2) is a semi-open form with a closed tendency and that H5N1^HM^ is a short-linker strain, which is not experimentally defined in this form; we have modelled it with an H1N1 type template (semi-open with closed tendency). By superimposing the structures at the beginning and at the end of the simulation, we can observe an approximation of the ED domains in the semi-open form. The movements of the two EDs are even more considerable in the open conformations, with average RMSD values (ED fit to RBD) of 23.1 and 20.4 Å for H1N1^HM^ and H5N1^XR^, respectively (Table 1, line RMSD ED fit to RBD, and Figure 5, green curves). While all three structures consistently remain in the open conformation, these motions include rotations about several alpha carbons of the linker region (Figure 4 and Figure 5).

The RMSD of each of the RBD and ED domains shows that the folding is very stable throughout the dynamics simulation (Table 1, Appendix A). This stability is confirmed on extended dynamics (340 ns, 298 ns, 180 ns) of three forms of H6N6 (closed, semi-open and open respectively) (Appendix A, Appendix A).

The RMSD results suggest that the NS1 protein is able to adopt open and closed forms regardless of the strain or the length of the linker. The semi-open form seems to be particular because it depends on the strain and the intermediate states of its initial structure (semi-open with closed or open tendency). Depending on the tendency of its initial semi-open conformation, it converges towards a more open conformation (H6N6^XR^) or more closed conformation (H1N1^XR^ or H5N1^HM^).

#### 2.2.2. Identification of Flexible Regions in the NS1 Structures

For each of the RBD and ED domains (including the linker region), we plotted the root mean square fluctuation of the alpha carbons (C𝛼RMSF) in the A and B chains in order to assess the flexibility regions. Our analysis revealed a region of high flexibility around residue 30 on the RBD domain, with RMSF values that can be higher than 1 Å. This region corresponds to the region of the α1-α2 connecting loop and to the N-terminal part of the α2 helix (Figure 1b and Appendix A). This flexibility is even more pronounced in open forms. In contrast, the most stable region is observed at the α1 helix, which is buried in the center of the RBD and constrained by its interactions with the α2, 2’ and α3, 3’ helices.

Beyond the linker region, which is known to be highly flexible, the ED domain also shows some flexible regions, particularly around residue 175, a flexible region corresponding to the N-terminal half of the α5 helix. This region is exposed to the solvent in both the open and the closed form (Appendix A).

### 2.3. Dynamic Motions of the ED Domains during Simulations

The RMSD results reflect the dynamic motions of the two EDs during the simulation. However, these movements of the EDs can be both related to opening or closing over time (translational motion) or reorientation (rotational motion). To support this result, the distances between the geometrical centers of the ED and RBD were computed during each of the nine simulations (Table 1) and compared to their initial values (Appendix A). These distances were plotted in Figure 6.

In the closed form, the d1 distance between the two ED domains is stable throughout the dynamics regardless of the strain, with the value ranging from 25 to 29 Å on average (Figure 6a, black curves). The EDs in the open form get closer, moving from 69–75 Å for the initial structure to 64–66 Å on average during the simulation (they are also closer to the RBD domain) (Figure 6a, green curves). Similarly for the semi-open forms, for the H1N1^XR^ and H5N1^HM^ structures (semi-open forms with a closed tendency), we observed a closing of the ED domains moving from 43–47 Å to 32–36 Å on average during the simulation and thus a closing, like an attraction, between the two domains (Figure 6a, red curves). For the H6N6^XR^ (semi-open form with an open tendency), the EDs maintain a constant distance throughout the trajectory. The distance value is around 60 Å on average and the EDs seem to be too far away to be able to approach each other and no force is exerted to promote this.

In addition to the distance d1 between the two ED monomers, we calculated the distance between each ED monomer and the RBD domain dimer during molecular dynamics (d2 and d3) (Figure 6b,c).

This analysis allowed us to identify an asymmetry in the movement of the ED domain monomers A and B in relation to the RBD, as in the case of the semi-open H1N1^XR^ structure where we could observe an approach of the ED monomers towards the RBD. The values d2 and d3 decreased from 49 and 46 Å (Table 1) during the molecular dynamics to an average distance of 42 and 45 Å, respectively. This asymmetry was also observed for the semi-open H5N1^HM^, where a distance of ED chain A and RBD (d2) increased from 35 Å at t = 0 ns to an average of 41.1 Å during the trajectory. This difference was not observed in d3 (Figure 6c, Table 1).

## 3. Discussion

This dynamic in silico work is the first to be performed on full length structures of the NS1 protein. Our objective in this study was to characterize the structural polymorphism by exploring three different strains in terms of sequence and shape. Based on the results of this study, we showed that NS1 in the closed form exhibits high stability in molecular dynamics simulations with a stable ED–ED dimerization interface that involves the strands of the beta-sheets. These closed-form structures are all derived from a homology-based model constructed from the only complete closed-form structure of NS1 available by crystallography. This structure corresponds to a closed-form obtained from the H6N6 strain for which a deletion of linker residues 80–84 was made (H6N6 short linker closed form, pdb id 4OPA) [15]. The authors hypothesized that this closed form may be characteristic of short linker strains. Our results suggest that the closed form is also possible and stable for long linker NS1 structures, as in the case of H1N1 or H6N6 strains. The closed form is characterized by a distance of about 25–28 Å between the geometric centers of the ED domains and a distance around 42 Å between the EDs and the RBD dimer. The folding of each ED domain and RBD dimers are very stable as shown by the RMSD and RMSF values. The dimerization interface is essentially maintained by contacts between the short β1 strand (residues 88–91) and linker residues and between the 𝛼4 helix and the loop connecting strand β6 and 𝛼5 helix. The ED–ED dimerization interface is a strand–strand interface as described by Kerry and colleagues [14]. The presence of a long linker shows that the dimeric ED in closed form can be moved from its initial position without affecting the closed form (maintaining the d1 distance throughout the dynamics) as observed for the H6N6 strain.

Structures of NS1 in the open conformation were all obtained from crystals formed by NS1 variants harboring a short linker that was either characteristic of the strain (H5N1 isolates, structure 3F5T in pdb) or genetically engineered (H6N6, pdb id 6OQE). In our study, we constructed long linker open-form homology models for the H1N1 and H6N6 variant based on these SL crystallographic structures as templates. The three NS1 strains with either LL or SL also appear to be stable in open form. Indeed, the RMSD curves reached a plateau, showing that the conformation remains globally stable after the initial structure relaxes in an aqueous environment. Similarly the distance d1, which measures the distance between the two EDs, was maintained throughout the dynamics. Thus, the three structures remain stable in open form, even if each ED can drift away from its initial position during the simulation. In this open form, the face of the beta-sheet with the short β1 strand of chain A ED domain is oriented toward the 𝛼3 helix of chain B RBD domain (and vice versa). For the H5N1 strain, we could observe in the superposition of structures at the beginning and at the end of the simulation (Figure 4) that one of the EDs showed a slight translation with respect to its initial position but also a significantly more twisted folding of the beta-sandwich. This was also described in the work of Bornholdt, et al., 2008 [20]. The authors suggested that this is due to either a constraint imposed by the presence of the RBD dimer or to the length of the linker in H5N1 (SL). Indeed, supporting this view, the twisting of the β-sandwich is less pronounced in the case of H6N6 and H1N1, both variants harboring a long linker.

In both the open and closed forms, we could observe that the face containing the short strand β1 (residues 88–91) is consistently less exposed to the solvent because it is either at the ED–ED interface (closed form), or at the ED–RBD interface (open form). This allows the 𝛼5 helix of the ED to be completely exposed to the solvent, including W187 residue that is localized at the C-terminal extremity of the helix. Since this helix (residues 170–188) and its residue W187 play a central role in another mode of ED-dimerization [14,17,18], these conformations are both expected to be prone to a multimeric state of the NS1 protein in solution [22,24].

The semi-open form is comprised of several states from very open to almost closed. In this conformation, both interaction sides (short strand β1 side or 𝛼5 helix side) are solvent-exposed, allowing several modes of interactions with various protein partners. Thus, it is a semi-open conformation that was crystal-captured in the interaction of H1N1 NS1 with TRIM25 [21]. In our molecular dynamic simulations, this is the form that underwent the largest conformational changes, exploring various degrees of the opening states (from very open to almost closed). While the H6N6^XR^ structure remained relatively stable in a conformation that tended toward the open state with a consistently high inter-ED distance, the EDs of the H1N1^XR^ structure underwent large motions, and the same was true for H5N1^HM^ that was built on the H1N1^RX^ template, even in spite of its short linker region. We hypothesize that this large conformational change corresponds to a relaxation of NS1 in freeing itself from the constraint imposed by its interaction with TRIM25.

Finally, our results show that the dynamics of the RBD dimer are extremely stable regardless of sequence variations, as indicated by the RMSD values calculated on this domain. The RMSF show a region that fluctuates more around position 30 (Appendix A). These results are in agreement with the work of Abi Hussein et al., 2020 [25]. This region coincides with the loop that connects the 𝛼1 and 𝛼2 helices. This loop is exposed to the solvent and, in the open form, faces an ED domain. Each ED has a very stable fold in all nine simulations, with slight variations, among which is twisting in the open form of the H5N1, which has a short linker. The RMSF values reveal several regions that present wider fluctuations, such as in the α5 helix region or in the β4–β5 strands region of the ED domain. These regions were also highlighted from RMSF calculations on a molecular dynamic simulation of an ED dimer from H1N1 strain [26]. In this study, the structure of the ED dimer was resolved by NMR experimental technique.

This in silico work confirms the compatibility of the three forms for the three strains, in agreement with the current full-length crystallography data [14,18,21,24]. Approaches such as NMR on full-length structures will allow in vitro confirmation of the NS1′s flexibility.

## 4. Materials and Methods

The three selected strains were H1N1, H5N1 and H6N6. The H1N1 strain was available in the PDB (Protein Data Bank) in full length and homodimeric structure in semi-open form with a long linker (pdb code: 5NT2) [21], H6N6 in full length and monomer structure in semi open form with long linker (pdb code: 4OPH) [13], and H5N1 structure in open form with short linker (del 80–84) (pdb code: 3F5T) [20]. The NS1 Protein sequences were extracted from the UniProt database and compared by a multiple alignment algorithm using Clustal Ω [27].

### 4.1. Preparation of the Crystal Structures

The structures of NS1 from H5N1 and from H6N6 recovered in the PDB were completed in term of the missing residues and dimers were rebuilt with the Protein Interfaces, Surfaces and Assemblies tool (PDBePISA v1.52, 2014) (https://www.ebi.ac.uk/pdbe/pisa/ accessed on 30 August 2021).

The structure of 5NT2 was already complete and in dimer form, we just dissociated it from its complex with TRIM25. The R38A/K41A engineered mutations for the three strains and also the W187A mutation specific to 5NT2 were reversed. The overlaid before and after reconstruction structures are illustrated with Pymol software (Figure 7).

### 4.2. Building Structural Models by Homology

For each strain, we built models by homology for the two complementary forms not known experimentally by crystallography, in order to study the behavior of the protein in the three strains and the three forms (closed, semi-open, open).

The closed form was modelled for the three strains on the 4OPA Xray template, which is a H6N6 structure where a deletion in the linker was induced (del 80–84). The open form was modeled for H6N6 and H1N1 strains on the 6OQE Xray template, which is also a H6N6 structure with a short linker. For these two forms, the missing residues in the linker were reversed in H6N6 and H1N1 models. The semi-open form was modeled for the H5N1 strain on the 5NT2 Xray template, reversing the deleted residues in the linker. As for crystal structures, the R38A/K41A mutations and also the W187A mutation specific to 5NT2 were reversed in all the models.

Table 2 summarizes the homology modeling steps performed with the MODELLER tool (v 9.23) [28].

### 4.3. Molecular Dynamics (MD) Simulation

The resulting nine structures described in Table 2, were first processed with ProPka to assign protonation states at pH 7 and to construct the missing side chains, ensuring that the new atoms were not reconstructed too close to existing atoms. MD simulations were performed with Gromacs 2019.5 [29], using the Amber99SB force field [30] under periodic boundary conditions. All structures were simulated as immersed in a cubic water box of the TIP3P water molecule model. Non-bonded interactions were truncated in a cut-off distance of 14 Å (for the closed and semi-open forms) and 18 Å (for the open form) for the electrostatic twin-range cut-off and the Van der Waals cut-off. The energy of the system was minimized over 50,000 cycles, using the steepest descent algorithm for energy minimization. Then, counter-ions were added to neutralize the system.

Each MD simulation was preceded by a 1ns heating/equilibration period during which harmonic constraints were imposed on the atomic positions of the protein and counter-ions. Each simulation was performed at constant temperature (300 K) and pressure (1 atm), the isothermal-isobaric ensemble (NPT), coupling the system to a heat bath, using the Berendsen algorithm. The LINCS algorithm was applied to all bond lengths to constrain them, allowing for a 2 fs integration time step. 150 ns simulations were carried out for each of the nine structures (3 forms * 3 strains). Dynamics of three forms of H6N6 were extended to check stability. The MD trajectories were visualized using Visualized Molecular Dynamics (VMD 1.9.2) [31] and the figures of the different snapshots were made with Pymol software [32].

Using the GROMACS tools, several properties were analyzed throughout the simulations to validate their quality and stability, including Root Mean Square Deviation (RMSD) and Root Mean Square Fluctuation (RMSF) calculated on the atomic coordinates of the C𝛼 atoms.

The C𝛼 RMSD analyses allowed us to study the stability of the whole protein but also of the RBD domain as a first step, and then the behavior of the ED domain in different strains and forms. The (C𝛼 RSMF) was analyzed to identify flexible residuals along the NS1 sequence.

### 4.4. Distance Calculations between Domains

In order to quantify the opening and closing movements of the ED domains, we decided to measure three distances between the geometric centers of the different domains (see Figure 8). We measured the distance between the geometric centers of the two monomers of the ED domain (composed of residues 86–203 for each chain); this distance was called d1. Then, we measured the distances between the geometric center of the RBD (composed of residues 1–72 for each chain A and B) and each geometric center of the ED monomers; these distances were respectively called d2 and d3. To perform these calculations, we used the “distance” module of the GROMACS software by creating indexes that include the atoms of each group (ED chain A, ED chain B, RBD dimer) and plotted the distance between the different groups during trajectory.

## 5. Conclusions

In this paper, we studied the dynamic properties of nine structural conformations of the NS1 protein. These nine conformations are both representative of different viral strains (H1N1, H6N6 and H5N1) and of the structural polymorphism described in the literature. To carry out this study, we combined both homology modeling approaches to complete the structural forms not available in the Protein Data Bank and molecular dynamics simulations to observe the stability and intrinsic properties of the NS1 protein as a function of its strain when its initial structure is imposed in one of the three structural forms (closed, semi-open, open).

Our results are consistent with the hypothesis that all three forms would be possible regardless of strain. However, we note differences in stability depending on the shape and degree of openness of the initial conformation. Closed forms are very stable. In the open forms, an asymmetry in the behavior of the EDs can be observed (particularly in the H5N1 strain) despite the maintenance of the form in its open state. This would be caused by a translation of the ED domain with respect to the RBD dimer and by a deformation of the β-sheet, which could be more twisted when the structure contains a short linker. The semi-open form is more unstable and has several intermediate states. Depending on the initial semi-open state, the shape tends to close or remain stable with the same degree of openness as the initial structure.

Homology and computational molecular dynamics approaches allow us to characterize the movements of the ED-RBD during simulations and give us a better understanding of the dynamic properties of the NS1 protein structure in its different forms and strains. This allowed us to conclude that the forms adopted by NS1 are not strain-dependent as NS1 is able to be in all three states regardless of the strain. This information is crucial in the development of an effective therapeutic strategy on the different strains, for instance, to limit interactions at the level of the RBD–RNA interface in order to prevent viral RNA replication, but also at the level of the ED domain interacting with different protein partners depending on the orientation of the ED monomers and the interface exposed to the solvent.

From the perspective of targeting NS1 for novel antiviral strategies, a first step towards the identification of small compounds targeting NS1 RBD was proposed by Abi Hussein et al., 2020 [25]. They explored the druggability of RBD using molecular dynamics simulations of one H6N6 crystal structure. They confirmed the remarkable stability of the H6N6 RBD structure and were able to identify potential binding pockets in the groove delimited by the antiparallel α2-helices that make up its RNA-binding interface. They highlighted the druggability of some of these pockets and the strict conservation of the residues involved, across the large sequence diversity of NS1, thus emphasizing the robustness of such an approach to identify broadly active RBD NS1-targeting compounds. In the present study, the molecular dynamics allowed us to study the stability of the RBD and characterize the ED-RBD movements during the simulations for different strains, thus providing a better understanding of the dynamic properties of the NS1 structure in its different forms. Our data confirm the remarkable stability of the RBD for different strains regardless of the conformation and of the linker length, thus emphasizing the interest of targeting the RBD by drug design approaches to develop a strain-independent therapy. Towards this aim, the next step will consist in extracting RBD druggable pockets, common to the different strains and then to use docking approaches to search for candidate compounds that bind these pockets.

Moreover, a study will be carried out to simulate models of these three strains built with a short linker versus a long linker in order to better understand and confirm certain hypotheses on the impact of the length of the linker region on the polymorphism of the NS1 protein, or to study other strains.

## Figures and Tables

**Figure 1 ijms-23-01805-f001:**
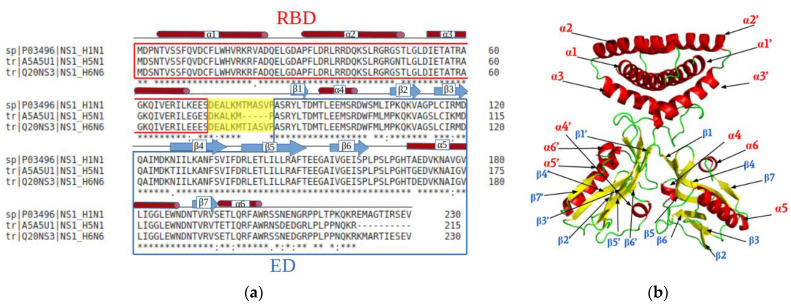
Multiple sequence alignment (Clustal Ω) of the three strains (H1N1, H5N1, H6N6) and secondary structure of the NS1 protein. (**a**) The residues boxed in red correspond to the RBD dimer (composed mainly of helices), the residues boxed in blue correspond to the ED domain (composed of α helices and β sheets) and the twelve residues boxed in yellow compose the linker region, which connects the RBD and ED domains. In the H5N1 sequence, the linker region is shorter due to five missing residues at positions (80–84). (**b**) Illustration in cartoon of the secondary structure (α helices in red, β strands in yellow and loop in green) of the NS1 protein (Pymol), pdb code: 4OPA.

**Figure 2 ijms-23-01805-f002:**
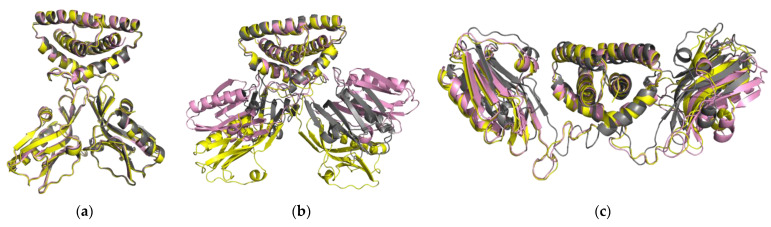
Illustration of the superposition of the structures of the H1N1 strains in yellow, H5N1 in grey and H6N6 in pink with Pymol. (**a**) Superposition of the closed forms of the three strains (H1N1^HM^, H5N1^HM^, H6N6^HM^); (**b**) superposition of the semi-open form structures of the three strains H1N1^XR^, H5N1^HM^, H6N6^XR^); (**c**) superposition of the open form structures (H1N1^HM^, H5N1^XR^, H6N6^HM^).

**Figure 3 ijms-23-01805-f003:**
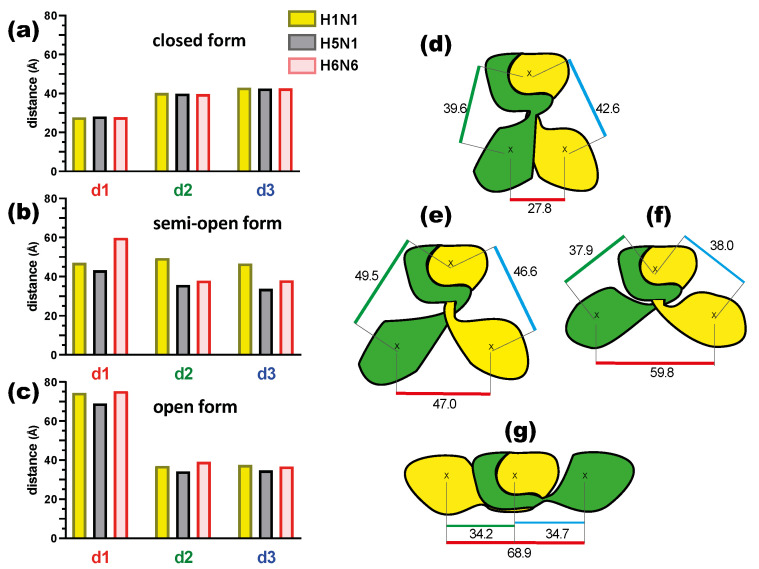
On the left are histograms of the three distances (d1, d2, d3) for the different strains (H1N1 in yellow, H5N1 in grey and H6N6 in pink): (**a**) in the closed form, (**b**) in the semi-open form and (**c**) in the open form. On the right, schematic representation of the dimeric NS1 protein (each chain is respectively in green and yellow) with the values of the distances d1, d2 and d3 indicated in Angstroms (underlined in red, green and blue respectively): (**d**) closed form, (**e**) semi-open form for H1N1, (**f**) semi-open form for H6N6 and (**g**) open form.

**Figure 4 ijms-23-01805-f004:**
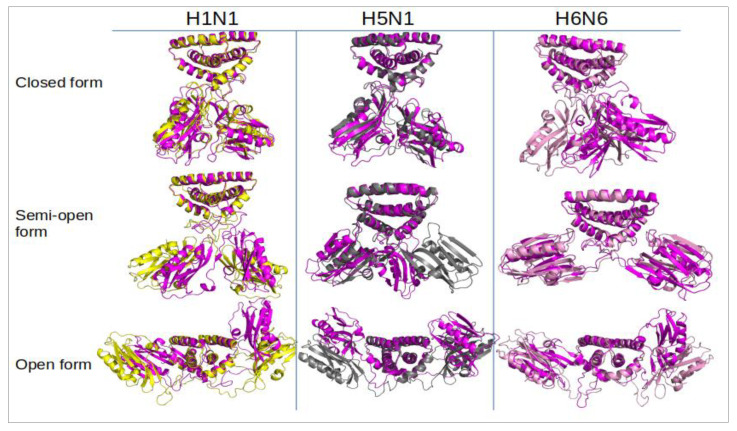
Superimposition of structures according to the three strains represented in the cartoon with Pymol. For each of the nine structures, the conformation at the end of the simulation (t = 150 ns) was extracted and superimposed on the initial structure (t = 0 ns). The initial H1N1, H5N1 and H6N6 conformations are colored respectively in yellow, grey and pink. The final conformations are shown in magenta.

**Figure 5 ijms-23-01805-f005:**
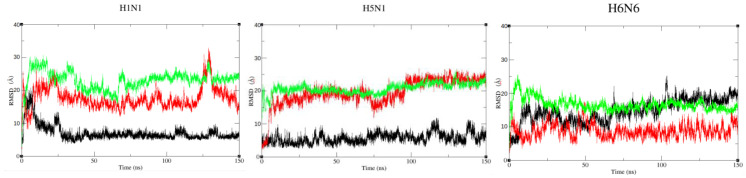
C𝛼 RMSD calculated on the ED domain fitting to the RBD for the three forms (closed, semi-open, open) respectively colored in black, red and green.

**Figure 6 ijms-23-01805-f006:**
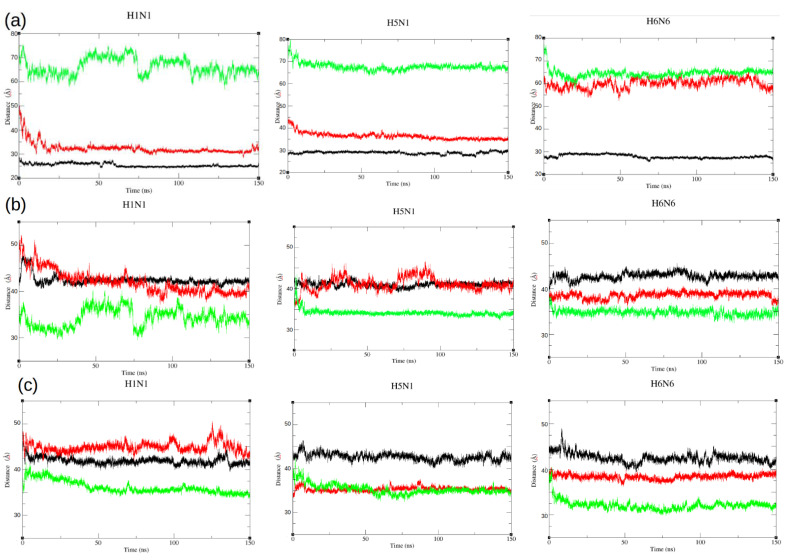
Average distances between geometric centers of ED domains for the three strains (H1N1, H5N1, H6N6) in the three forms (closed, semi-open and open), respectively, colored in black, red and green. (**a**) Distances (d1) between geometric centers of ED domains; (**b**) Distances (d2) between the geometric center of ED domain chain A and the geometric center of RBD dimer; (**c**) Distances (d3) between the geometric center of ED domain chain B and the geometric center of RBD dimer.

**Figure 7 ijms-23-01805-f007:**
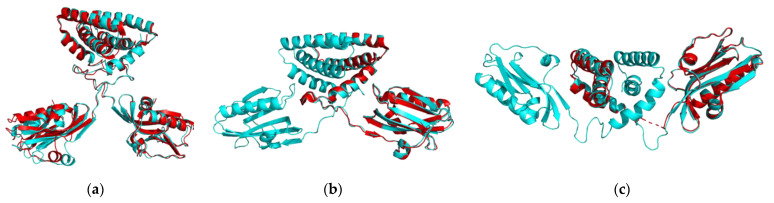
Alignment of structures before and after reconstruction with the PISA tool. The red representations are the crystallographic structures before reconstruction and the cyan representations are the reconstructed structures: (**a**) structure of the NS1 protein of the H1N1 strain in the semi-open form with closed tendency (5NT2); (**b**) structure of the NS1 protein of the H6N6 strain in the semi-open form with open tendency (4OPH); (**c**) structure of the NS1 protein of the H5N1 strain in the open form (3F5T).

**Figure 8 ijms-23-01805-f008:**
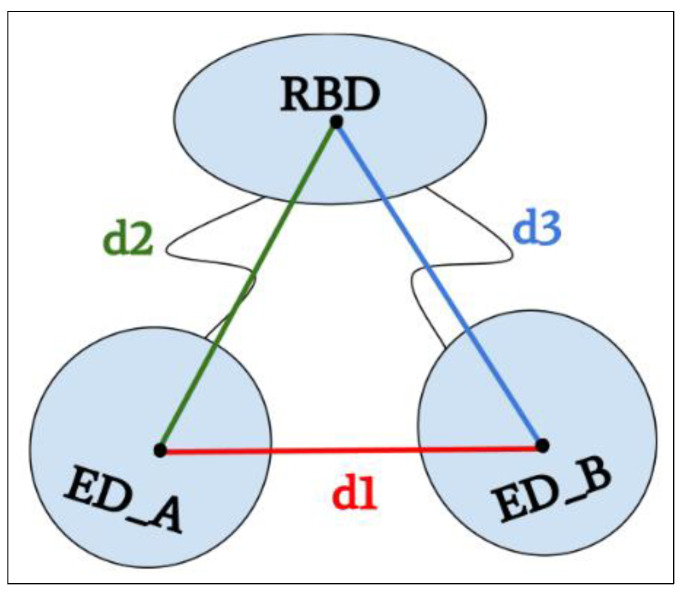
Distance diagrams (d1, d2, d3). The distance d1 is the distance between geometric centers of the ED monomers. The distances d2 and d3 are the distances between geometric centers of each monomer chain A and B of the ED domain and the RBD dimer.

**Table 1 ijms-23-01805-t001:** Average C𝛼 RMSD and distances for each strain (H1N1, H5N1, H6N6) in the three forms (closed, semi-open, open) during the trajectory. RMSD_All is the average RMSD calculated over the whole protein, RMSD_RBD is the RMSD of the RBD dimer after fitting to the RBD, RMSD ED_A and ED_B are the RMSD of each monomer A and B of the ED domain calculated independently and RMSD ED fit on RBD corresponds to the RMSD of the EDs relative to the reference frame attached to the RBD. The distances (d1, d2, d3) correspond respectively to the average distances over time between the two ED monomers, between ED chain A and RBD and between ED chain B and RBD. The average values are indicated with their standard deviation.

Strains/Forms	Closed Form			Semi-Open Form			Open Form		
Strains	H1N1^HM^	H5N1^HM^	H6N6^HM^	H1N1^XR^	H5N1^HM^	H6N6^XR^	H1N1^HM^	H5N1^XR^	H6N6^HM^
RMSD Values (in Å)									
RMSD All	3.5 ± 0.5	2.3 ± 0.4	5.0 ± 1.1	10.2 ± 1.0	8.6 ± 1.4	4.2 ± 0.7	10.3 ± 1.6	11.9 ± 2.1	8.9 ± 0.7.9
RMSD RBD	1.9 ± 0.1	1.9 ± 0.1	2.4 ± 0.2	3.3 ± 0.1	2.3 ± 0.1	2.2 ± 0.2	3.1 ± 0.01	3.0 ± 0.1	2.7 ± 0.1
RMSD ED_A	1.5 ± 0.1	1.2 ± 0.1	1.1 ± 0.1	1.2 ± 1	1.6 ± 0.2	1.3 ± 0.2	1.8 ± 0.2	2.5 ± 0.2	2.1 ± 0.1
RMSD ED_B	1.6 ± 0.1	1.2 ± 0.2	1.3 ± 0.1	1.2 ± 0.1	1.1 ± 0.1	1.1 ± 0.1	1.4 ± 0.1	1.9 ± 0.2	2.1 ± 0.1
RMSD ED fit to RBD	7.1 ± 2.4	5.6 ± 1.5	14.7 ± 3.4	17.6 ± 3.1	19.5 ± 4	8.8 ± 2.0	23.1 ± 2.7	20.4 ± 2.9	16.6 ± 1.8
Distances values									
Distance d1	25.4 ± 0.7	28.8 ± 0.6	27.8 ± 0.8	32.4 ± 2.3	36.4 ± 1.5	60.0 ± 2.0	66.3 ± 3.4	66.5 ± 1.7	64.5 ± 1.7
Distance d2	42.4 ± 1.0	40.9 ± 0.5	42.7 ± 0.7	42.0 ± 2.0	41.1 ± 1.6	38.5 ± 0.6	34.3 ± 1.9	36.3 ± 1.0	34.7 ± 0.6
Distance d3	41.9 ± 0.7	42.6 ± 0.8	42.4 ± 1.0	44.9 ± 1.0	35.3 ± 0.4	38.3 ± 0.5	36.1 ± 1.3	34.5 ± 1.3	32.0 ± 1.0

**Table 2 ijms-23-01805-t002:** The nine initial structures from crystallography experiment or homology modeling. For each structure, the sequence Uniprot identifier, the pdb id of the crystallographic structure or of the template structure used for homology modeling and the reversed mutations are listed.

Strains/Forms	Closed Form	Semi-Open Form	Open Form
H1N1	**H1N1^HM^**	**H1N1^XR^**	**H1N1^HM^**
	Uniprot identifier: **P03496**	Uniprot identifier: **P03496**	Uniprot identifier: **P03496**
	Xray structure template: 4OPA	Xray structure: 5NT2	Xray structure template: 6OQE
	Reverse mutations: R38A/K41A/W187A	Reverse mutations: R38A/K41A/W187A	Reverse mutations: R38A/K41A/W187A
H5N1	**H5N1^HM^**	**H5N1^HM^**	**H5N1^XR^**
	Uniprot identifier: **A5A5U1**	Uniprot identifier: **A5A5U1**	Uniprot identifier: **A5A5U1**
	Xray structure template: 4OPA	Xray structure template: 5NT2	Xray structure: 3F5T
	Reverse mutations: R38A/K41A	Reverse mutations: R38A/K41A	Reverse mutations: R38A/K41A
H6N6	**H6N6^HM^**	**H6N6^XR^**	**H6N6^HM^**
	Uniprot identifier: **Q20NS3**	Uniprot identifier: **Q20NS3**	Uniprot identifier: **Q20NS3**
	Xray structure template: 4OPA	Xray structure: 4OPH	Xray structure template: 6OQE
	Reverse mutations: R38A/K41A	Reverse mutations: R38A/K41A	Reverse mutations: R38A/K41A

## Data Availability

Data is contained within the article.

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
