# Peer review of "Influenza A Virus NS1 Protein Structural Flexibility Analysis According to Its Structural Polymorphism Using Computational Approaches"

_ijms, 2022, doi:10.3390/ijms23031805_

Round 1

Reviewer 1 Report

The manuscript, "Influenza A virus NS1 protein structural flexibility analysis according to its structural polymorphism using computational approaches," by Naceri et al., is a well executed and well written study of a particular region of several strains of flu virus proteins. Introduction and discussion and analysis are in-depth and thorough. My only real issue with the manuscript is that while the authors discuss the potential use of this region of the protein as a drug target, their analysis falls short of taking that step. Thus, the level of interest for this manuscript would be less than if they had followed through with further analysis related to drug design.

A few small corrections:

Line 65: should be “is comprised of”

Line 141: modelized should be modelled

Lines 143-144: the sentence presented across these lines does not make sense.

Reviewer 2 Report

The authors in this manuscript used computational approaches to analyze influenza A virus NS1 protein structural flexibility. They also used their NS1 structural polymorphism to in silico validate the dynamic properties and the flexibility. These positive findings improve the scientific reliability of this study, but no functional assays in vitro and in vivo further prove their modeling data. Although there is shortcoming, the manuscript is still publishable in this journal.

Reviewer 3 Report

The manuscript reports an interesting computational study aiming to achieve insights into the polymorphism and flexibility of the molecular architecture of Influenza A virus’s non structural proteins. The quality of the presented text is acceptable, the language is clear, and results as well the applied methodology are reported in a quite good comprehensive way. 
Despite these aspects, the paper shows some pitfalls most likely related to the short length of the dynamics runs that in the contest of the study represent the most relevant step. In my opinion 150 ns is not a sufficient duration to explore any evolutionary conformational changes that indeed represent relevant three-dimensional features as well as distinguishing the diverse NS1 protein herein stuided. Nowadays, 500-1000 ns dynamics can be easily carried out on standard graphic cards and GPU within acceptable time and without  dramatical economic efforts
Authors are then encouraged to review their studies extending where feasible the trajectory lengths to further explore the dynamical behaviour of the selected molecules
At the same time, minor changes are need:
1. Data reported Table 1 and 2 will be better perceived if shown as histograms and not numbers
2. Report distance values in Ang and not nm

Round 2

Reviewer 3 Report

The authors solved all the highlighted critics and pitfalls of their manuscript that it is now merit to be published